# Overexpression of the Melatonin Synthesis-Related Gene *SlCOMT1* Improves the Resistance of Tomato to Salt Stress

**DOI:** 10.3390/molecules24081514

**Published:** 2019-04-17

**Authors:** Dan-Dan Liu, Xiao-Shuai Sun, Lin Liu, Hong-Di Shi, Sui-Yun Chen, Da-Ke Zhao

**Affiliations:** 1School of Agriculture, Yunnan University, Kunming 650091, China; liudandan@ynu.edu.cn (D.-D.L.); sunxiaoshuai2019@163.com (X.-S.S.); qiuqiu12-09@163.com (L.L.); dong19850412@163.com (H.-D.S.); 2Biocontrol Engineering Research Center of Plant Disease & Pest, Yunnan University, Kunming 650504, China; 3Biocontrol Engineering Research Center of Crop Disease & Pest, Yunnan University, Kunming 650504, China; 4School of Life Science, Yunnan University, Kunming 650504, China

**Keywords:** tomato, *SlCOMT1*, melatonin, genetical transformation, salt stress

## Abstract

Melatonin can increase plant resistance to stress, and exogenous melatonin has been reported to promote stress resistance in plants. In this study, a melatonin biosynthesis-related *SlCOMT1* gene was cloned from tomato (*Solanum lycopersicum* Mill. cv. *Ailsa Craig*), which is highly expressed in fruits compared with other organs. The protein was found to locate in the cytoplasm. Melatonin content in *SlCOMT1* overexpression transgenic tomato plants was significantly higher than that in wild-type plants. Under 800 mM NaCl stress, the transcript level of *SlCOMT1* in tomato leaf was positively related to the melatonin contents. Furthermore, compared with that in wild-type plants, levels of superoxide and hydrogen peroxide were lower while the content of proline was higher in *SlCOMT1* transgenic tomatoes. Therefore, *SlCOMT1* was closely associated with melatonin biosynthesis confers the significant salt tolerance, providing a clue to cope with the growing global problem of salination in agricultural production.

## 1. Introduction

Melatonin (*N*-acetyl-5-methoxytryptamine) is also known as the pineal hormone, because it was first detected in the pineal gland of cattle by Lerner and colleagues in 1958 [1]. Since then, melatonin has been reported to regulate important physiological processes in mammals [2,3,4,5], such as circadian rhythm, mood, sleep, body temperature, activity, food intake, sexual behavior, and seasonal reproduction [6]. In addition to its physiochemical functions in mammals, melatonin also plays important roles in plant physiology [7]. Melatonin from plants, i.e., ‘phytomelatonin’, was originally regarded as an endogenous antioxidant molecule [8]. Based upon its powerful ROS scavenging activity, its roles in plant development as well as the resistance of plants to biotic and abiotic stress are also recognized. In detail, various physiological functions of melatonin in plants have been discovered, including promotion of explants growth, formation of the rhizome, induction of leaf senescence, and regulation of flowering, photosynthesis, circadian rhythms, and seed germination [9,10,11,12,13,14,15,16,17]. As mentioned above, phytomelatonin itself shows an effective antioxidative property, and it could clear the ROS generated under different kinds of abiotic stresses such as chemical pollution, ultraviolet radiation, herbicides, drought, heat, cold, and salinity, thus enhancing the abiotic resistances for plants [18,19,20,21].

The synthesis of melatonin is accomplished through four main reactions involving at least six enzymes, namely, tryptophan hydroxylase (TPH, EC 1.14.16.4), tryptophan decarboxylase (TDC, EC 4.1.1.28), tryptamine 5-hydroxylase (T5H, EC 1.1.13), serotonin *N*-acetyltransferase (SNAT, EC 2.3.1.87), *N*-acetylserotonin-*O*-methyltransferase (ASMT, EC 2.1.1.4), and caffeic acid *O*-methyl-transferase (COMT, EC 2.1.1.68) [22]. Tryptamine is produced by TDC in the cytoplasm, followed by serotonin generation in the endoplasmic reticulum [23]. Afterwards, serotonin is converted into *N*-acetylserotonin in the chloroplast and 5-methoxytryptamine in the cytoplasm by SNAT and ASMT, through acetylation and methylation, respectively. Then, these two intermediates are converted to melatonin by ASMT in the cytoplasm or SNAT in the chloroplast [24,25]. Similar to ASMT, COMT has been reported to play a pivotal role in the synthesis of phytomelatonin, specifically existing in the plant cytoplasm [26]. In *Arabidopsis*, serotonin can be converted to 5-methoxytryptamine by COMT methylation in the third step of melatonin synthesis, and then catalyzed by SNAT to melatonin [27]. On the basis of the type of enzymatic catalysis, COMT also belongs to the O-methyltransferase (OMT) family. The OMT family converts methylation sites on *S*-adenosyl-l-methionine into various secondary metabolites, including flavonoids, phenyl- propanoids, and alkaloids [28]. Some plants, including dicots, lack ASMT homologs, suggesting that COMT plays an important role in the last step of melatonin synthesis [26,27]. In *Arabidopsis*, the catalytic activity of COMT is much higher than that of ASMT during the synthesis of melatonin, and the melatonin content is significantly reduced in mutant *Arabidopsis* when the *COMT* gene is silenced [26]. In plants, *COMT* gene improves melatonin production and positively contributes to strengthen both biotic and abiotic stress resistance in plants [29,30]. 

Tomato, one of the most highly consumed and extremely important horticultural plants, has been studied as a model plant for some aspects of plant growth and development [31]. Thus, it represents an ideal model organism to study the melatonin synthesis pathway. In recent years, several investigators have studied the effects of exogenous melatonin on tomato plants exposed to abiotic stress. For example, supplying additional melatonin improved the resistance to cadmium and water deficit in tomatoes [32,33]. Furthermore, exogenous melatonin promoted root development by regulating auxin and nitric oxide signaling in tomato [34]. In our study, *SlCOMT1*, a gene related to the biosynthesis of melatonin, was isolated from tomato. Phylogenetic relationships, subcellular localization and temporal-spatial expression were analyzed. To further characterize its potential stress-tolerant function in tomato, transgenic tomato plant with exogenous *SlCOMT1* was generated and analyzed. Furthermore, the relationship between melatonin production and salt resistance of transgenic tomato plants was investigated.

## 2. Results

### 2.1. Molecular Cloning and Sequence Analysis of SlCOMT1

To investigate the function of the melatonin biosynthesis-related gene *COMT* in tomato, *AtCOMT* gene sequence from *Arabidopsis* was used to search the tomato genome. Five similar sequences were identified, which were then aligned using an online tool (http//:www.ncbi.nlm.nih.gov/). The results show that the SlCOMT1 protein contains a dimerization domain (11–62aa) at the 5′ end and an adomet-MTase domain (105aa–321aa) at the 3′ end (Figure 1A). Phylogenetic tree analysis revealed protein homology between SlCOMT1 and AtCOMT (Figure 1B), showing that SlCOMT1 is the closest homolog of AtCOMT in tomato, with amino acid sequence similarity between them at 69.29%. Therefore, the gene was identified and designated as *SlCOMT1* (LOC101251452) (Figure 1C).

To obtain the *SlCOMT1* gene from tomato, specific *SlCOMT1*-F/R primers were used for PCR (Appendix A), and a band of expected size was detected on a 1.2% agarose gel (Appendix A), which was sequenced and characterized. The full-length *SlCOMT1* cDNA is 1023 bp, encoding a protein of 341 amino acids with a molecular weight of 37 kD and with an isoelectric point of 5.74. The *SlCOMT1*-PET32a recombinant vector was constructed to induce expression of the SlCOMT1 protein. The result shows that the SlCOMT1 protein is approximately 54 kD with a His-tag which is in line with our prediction (Figure 2).

Compared with COMT proteins in other plants, the SlCOMT1 protein sequence shows various conserved functional regions (Figure 3A). These proteins share five structurally conservative domains, including VVDVGGGTG, EHVGGDMF, GINFDLPHV, GGKERT, and NGKVI (Figure 3B), indicating that the sequence and function of SlCOMT1 are similar to these of other plant COMT proteins.

### 2.2. Phylogenetic Tree Analysis of SlCOMT1

To reveal the relationship between the SlCOMT1 protein and other plant COMT proteins, the phylogenetic tree of SlCOMT1 and other COMT proteins was constructed using MEGA 5.0 (Figure 4). The results show that SlCOMT1 and *S. tuberosum* COMT protein are classified into one category, suggesting that SlCOMT1 and StCOMT are derived from a recent ancestor, since reported in two congeneric species.

### 2.3. Structure Prediction of SlCOMT1 Protein

The SOPMA online tool (https://npsa-prabi.ibcp.fr/cgi-bin/secpred_sopma.pl) was used to predict the secondary structure of the SlCOMT1 protein. The results suggest that the secondary structure of the protein is mainly composed of four parts, of which α-helices account for 46.2%, followed by random coils (29.24%), extended strands (16.08%), and β-turns (8.48%) (Figure 5A). Additionally, a three-dimensional structure of the SlCOMT1 protein was constructed to verify the above results using the Phyre 2 online tool (http://sbg.bio.ic.ac.phyre/) (Figure 5B).

### 2.4. SlCOMT1 Protein Subcellularly Localized in the Cytoplasm

The subcellular localization of SlCOMT1 in tobacco leaves was determined using a chimeric SlCOMT1-GFP fusion protein and a transient transfection assay. Green fluorescence was observed in the cytoplasm of epidermal cells transfected with the *35S:SlCOMT1-GFP* plasmid (Figure 6), revealing that SlCOMT1 is localized in the cytoplasm.

### 2.5. Temporal and Spatial Expression of SlCOMT1

The temporal and spatial expression of *SlCOMT1* was investigated by real-time PCR. The results show that *SlCOMT1* is constitutively expressed in tomato tissues, including roots, shoots, leaves, flowers, and fruits, with variable expression levels in these tissues. The lowest expression is in roots and the highest expression is in fruits (Figure 7), suggesting that *SlCOMT1* might be involved in the regulation of fruit development.

### 2.6. SlCOMT1 Overexpression Increased the Melatonin Content and Salt Resistance in Tomato

To characterize the function of the *SlCOMT1* gene in plants, *SlCOMT1* driven by the 35S promoter was genetically transformed into tomato to generate transgenic lines, and two with different transcript levels were used for further study, namely, OE-1 and OE-2 (Appendix A). In light of the role of COMT in the synthesis of melatonin, the melatonin content was measured in wild-type and transgenic tomato plants. The content of melatonin was higher (30–35 pg/mL) in overexpression transgenic plants, compared to 27.365 pg/mL in wild-type plants (Figure 8A), indicating that *SlCOMT1* functions in melatonin production. 

It was reported that exogenous melatonin could increase the resistance of plants to salt stress in apple [35]. In the current study, wild-type and transgenic tomato plants were treated using 800 mM NaCl. One week later, the leaves appeared droop in wild-type tomato and displayed wilting. In contrast, the leaves looked normal and healthy in transgenic plants (Figure 8B). Additionally, the levels of superoxide, hydrogen peroxide and proline were measured in both wild-type and transgenic plants. Under normal development, the levels of superoxide, hydrogen peroxide and proline varied slightly both in wild-type and transgenic tomato plants (Figure 9A–C). Compared with control plants, the levels of superoxide and hydrogen peroxide increased both in wild-type and transgenic tomato plants under treatment using 800 mM NaCl, but they were lower in transgenic plants (Figure 9A,B). On the contrary, the level of proline was significantly higher in transgenic plants than in control plants (Figure 9C). Thus, overexpression of *SlCOMT1* improves melatonin production and enhances salt tolerance in tomato.

## 3. Discussion 

A high concentration of salt in soil is one of the most serious abiotic stresses for plants [36]. Melatonin plays important roles in various mechanisms that protect plants from external environmental stresses [17,37]. In this study, five *COMT* homologous genes were identified in the tomato genome, and the melatonin synthesis-related gene *SlCOMT1* was isolated based on homology comparison using *Arabidopsis* AtCOMT protein and the five tomato COMT proteins. Furthermore, SlCOMT1 protein was localized in the cytoplasm, suggesting that it might catalyze serotonin into 5-methoxytryptamine in the cytoplasm.

During normal cellular metabolism, ROS are generated by oxidative reaction process of mitochondrial respiration and photosynthesis process, and they act as signaling molecules during cellular repair processes at low amounts [38]. Once the plant is under environment stresses, its cells simultaneously initiate a series of response mechanisms and stress signals, such as the activation of cellular ROS scavenging mechanisms, which can trigger the production of reactive oxygen scavenging enzymes and antioxidants, including POD and SOD which work on scavenging excessive ROS, thereby alleviating or eliminating oxidative stress [39]. Under salt stress, the dynamic equilibrium of the production and elimination of reactive oxygen species in plant cells is disrupted, thereby causing the production of superoxide. Therefore, onset of cellular oxidative damage is the hallmark of salt stress, which is indicated by levels of superoxide and hydrogen peroxide [40]. Furthermore, proline in small amount plays multiple roles, such as stabilization of membrane and proteins, redox homeostasis and regulation of salt stress-responsive genes expression [41,42]. Superoxide, hydrogen peroxide and proline contents can respond to many environmental stresses in plants, including salt stress, and the accumulation of hydrogen peroxide and superoxide can disrupt the dynamic balance of cells under environmental stress [43,44,45]. Exogenous melatonin could have helped the tomato plants to bear the environmental stress by regulating the antioxidant system, proline and carbohydrates metabolism [46]. In this study, under treatment using NaCl, *SlCOMT1* overexpression transgenic plants displayed the increased proline and the decreased hydrogen peroxide and superoxide levels, which were resulted from the reduced oxidative damage by extra melatonin that can scavenge ROS in plant cells. As a result, melatonin produced by the *SlCOMT1* overexpression improved the growth characteristics of tomato compared to wild-type plants.

There are many cases suggesting that exogenous melatonin plays important roles in plant development and abiotic stress tolerances. For instance, exogenous melatonin has been reported to promote seed germination and seedling growth, and regulate the expression of growth-related genes involved in cell wall growth and expansion [46]. The molecule could improve plant tolerance to alkaline stress, drought stress, Cd stress and salinity stress by improving photosynthesis activity [47]. In addition, exogenous melatonin could confer cold tolerance in cucumber seedling by upregulating the expression of *ZAT12* gene accompanied by higher endogenous polyamine accumulation and higher ROS clearance system activity [48]. In this study, the melatonin content was elevated with the increased expression level of *SlCOMT1* in tomato, indicating that the *SlCOMT1* gene was involved in the synthesis of melatonin. Additionally, *SlCOMT1* overexpression transgenic tomato plants enhanced the resistance to salt stress. Therefore, the results indicate that *SlCOMT1* may be a key factor in regulating the response of plants against abiotic stresses by elevating melatonin production, therefore enhancing resistance to abiotic stress in tomato and other plants.

In conclusion, the present study shows that melatonin biosynthesis-related gene *SlCOMT1* isolated from tomato is localized in the cytoplasm, and is highly expressed in fruits. Melatonin content in *SlCOMT1* overexpression transgenic tomato plants is significantly higher than that in wild-type plants. The transgenic plants display increased proline levels and decreased hydrogen peroxide and superoxide levels, and the transgenic tomatoes tolerated salt stress better than the wild-type tomatoes. The results indicate that *SlCOMT1* is closely relate to melatonin production and functions in the improvement of plant resistance to abiotic stress.

## 4. Materials and Methods

### 4.1. Plant Materials

Tomato (*S. lycopersicum* Mill. cv. *Ailsa Craig*) was used for generating transgenic plant. After the surface of the tomato seeds was sterilized and soaked, the seeds were placed on a wet filter paper in a petri dish in a dark environment at 28 °C for germination. The seeds were transferred to a seedling tray containing sand and peat (1:1). After leaf growth, the seedlings were transplanted to pots containing matrix culture.

### 4.2. Cloning and Homology Analysis of the SlCOMT1 Gene

RNA Plant Plus reagent (Tiangen, Beijing, China) was used for total RNA extraction from tomato leaves. The total RNA served as the template for the synthesis of cDNA using the PrimeScript first-strand cDNA synthesis kit (Takara, Dalian, China). Primers (Appendix A) were designed based on sequences downloaded from the tomato genome website, and PCR was carried out using cDNA as a template. The reaction volume was 50 μL (Appendix A). The reaction consisted of 35 cycles (Appendix A). The PCR products were separated by 1.2% agarose gel electrophoresis, and the desired band was recovered and ligated to the pMD18-T cloning vector for sequencing.

### 4.3. Bioinformatics Analysis of the SlCOMT1 Gene

The sequence of the amplified gene was used to identify its ORF by DNAStar Lasergene EditSeq (7.1.0, DNAStar, Madison, WI, USA), and then the nucleotide sequence was translated into an amino acid sequence by DNAMAN 6.0.3.99 software (Lynnon Biosoft, San Ramon, CA, USA). Nucleotide and amino acid sequence similarity alignments were performed by Blast (http://blast.ncbi.nlm.nih.gov/). The ProtParam protein analysis tool (http://web.expasy.org/protparam/) was used to analyze the molecular weight, theoretical isoelectric point, and other protein properties. The 3D model of the encoded protein was generated using the Phyre 2 online tool (http://sbg.bio.ic.ac.phyre 2/, London, England). The neighbor-joining phylogenetic tree of *SlCOMT1* was constructed using MEGA 5.0 software (Arizona State University, Tempe, AZ, USA).

### 4.4. Expression Analysis of the SlCOMT1 Gene

The BIO-RAD IQ5 (Bio-Rad, Hercules, CA, USA) was used for real-time PCR. The internal reference gene was 18S. All PCR reactions were performed three times. The reaction volume was 20 μL ( Appendix A). The real-time PCR reaction conditions were as follows: pre-denaturation at 95 °C for 10 min, followed by 40 cycles of denaturation at 95 °C for 15 s, annealing at 60 °C for 15 s, and extension at 60 °C for 45 s. The 2^−∆∆CT^ method was used for quantitative analysis.

### 4.5. Construction of the Prokaryotic Expression Vector

We designed specific restriction site primers based on the *SlCOMT1* gene sequence: *SlCOMT1*-Sma I–F (5′–CCCGGGAATGCAACTGGCGAGTGCC–3′, the underlined nucleotides correspond to the Sma I site) and *SlCOMT1*-Pst I-R (5′–CTGCAGAGAGATTCTTGGTGAATTCCA–3′, the underlined nucleotides correspond to the Pst I site) (Appendix A). The *SlCOMT1* pMD18-T plasmid served as the template for PCR, followed by product purification. In addition, the pET-32a expression vector and purified PCR product were digested with Sma I and Pst I, and the digested products were recovered. The recovered, digested products were ligated using *T4 ligase* at 16 °C overnight, and *E. coli* BL21 were transformed with the pET32a-*SlCOMT1* recombinant plasmid. Bacterial colonies were screened for ampicillin resistance. The positive clones were sent to Qingdao Qingke for sequencing and identification.

### 4.6. Induction of SlCOMT1 Protein Expression

To verify the substrate specificity of the SlCOMT1 protein, we introduced the *SlCOMT1* cDNA into the pET32a expression vector containing a His tag and expressed it in *E. coli*. We collected 1 mL of the bacterial solution, which served as the control. To the remaining bacterial solution, we added 1 mM IPTG, and after 6 h of induction at 37 °C, 1 mL of the bacterial solution was withdrawn, and the remaining bacterial solution was centrifuged to collect the bacterial cells. The bacterial cells were resuspended in 3 mL of 8 M urea and sonicated. The supernatant and precipitate were collected. After denaturing the precipitate, 1 mL of the solution was collected once again. All samples were subjected to SDS-PAGE.

### 4.7. Establishment of Genetically Modified Tomatoes

Firstly, construction of the SlCOMT1 expression vector was performed as follows. Specific restriction site primers were designed as shown in Appendix A. *SlCOMT1* was digested from the pMD18-T cloning vector and purified, followed by the digestion of the pCXSN-Myc vector with the same enzymes. Both fragments were ligated with *T4 ligase* at 16 °C for subsequent transformation into *E. coli*, and positive clones were identified. The *SlCOMT1-Myc* overexpression vector was transformed into *Agrobacterium* LBA4404 to obtain overexpressing (*OE-SlCOMT1*) transgenic tomato plants. Afterwards, *SlCOMT1* overexpression transgenic tomatoes were obtained as follows. Wild-type tomato seeds were sterilized with 70% ethanol, treated with 26% sodium hypochlorite, and then washed with sterile ddH_2_O 4–5 times. The seeds were placed in seed germination medium for cultivation. After one week, the seeds germinated and reached the cotyledon stage. The cotyledons were cut into leaf discs and stem segments (Appendix A), respectively, transferred to pre-culture medium with the incision side down (Appendix A), and cultured in a dark environment for two days. A single *agrobacterium* colony carrying the expression plasmid was cultured in LB medium supplemented with antibiotics (kanamycin and rifampicin) at 28 °C under constant agitation. When the OD_600_ reached 0.6, the bacterial cells were collected and suspended in MS medium. The pre-cultured explants were infected for 10 min, and the excess bacterial liquid on the surface of the explants was absorbed by a filter paper. The inoculated explants were placed on pre-culture medium and cultured in a dark environment at 28 °C for 1 day. Thereafter, the infected explants were placed on tomato differentiation medium (Appendix A) and cultured under normal conditions. When the adventitious buds grew into 2–3 cm seedlings (Appendix A), they were cut and transferred to rooting medium, with 3–4 strains per culture flask. The seedling roots (Appendix A) were washed, transplanted to pots containing vermiculite and perlite (1:1 ratio), and covered with a moisturizing film for 3–5 days. The moisturizing film was gradually removed to yield healthy tomato seedlings. Thereafter, the seedlings were transplanted for cultivation.

### 4.8. Subcellular Localization of the SlCOMT1

Confocal laser-scanning microscope (Zeiss LSM 510 META, Jena, Germany) was used to investigate the subcellular localization of SlCOMT1. Primers containing Sal I and Bam HI restriction sites (Appendix A) were used for PCR. The PCR product was gel purified and ligated into the 35S:PRI101-GFP vector. The expression plasmid was transformed into *Agrobacterium* 4401 by the freeze-thaw method, and then injected into two-week old tobacco leaves. After transient expression, images were acquired with a confocal microscope.

### 4.9. Molecular Identification of Transgenic Tomato

Transgenic tomatoes were performed by PCR and quantitative RT-PCR. To identify the transgenic tomatoes, the cDNA of both wild-type and transgenic plants were used as a template to detect the expression level of *SlCOMT1*, the plasmid DNA of *SlCOMT1* was used as positive control, and ddH_2_O as negative control. The transgenic samples with increased *SlCOMT1* expression level were used for further study. Transcript level of *SlCOMT1* was detected using quantitative RT-PCR. Primers for quantitative RT-PCR (Appendix A) were used to detect the expression level of *SlCOMT1*. Quantitative RT-PCR reaction conditions were listed as Appendix A, with 30 cycles for fluorescence collection from denaturation to extension, and finally making a quantitative analysis by 2^−∆∆CT^ method.

### 4.10. Measurement of Melatonin Content

Wild-type and transgenic tomato plants with uniform growth potential were weighed and 0.1 g of fresh leaves were used to determine the melatonin content. Melatonin content in the leaves was measured using an enzyme-linked immunosorbent assay (Shanghai Enzyme Biotechnology, Shanghai, China). The standard, blank, and sample wells were assayed individually, and the absorbance at 450 nm was measured. The standard curve was generated after measuring the standard product, and the wild-type and transgenic tomato plants were assayed individually.

### 4.11. NaCl Treatment of Transgenic Tomato Plants and Detection of Hydrogen Peroxide, Superoxide and Proline 

Wild-type and transgenic tomato seedlings with uniform growth potential were domesticated and then transplanted to medium containing matrix culture. After a period of healthy plant growth, we harvested 0.1 g of different plant leaves, and the contents of proline, hydrogen peroxide, and superoxide in the plants were determined. Two groups each made of both wild-type and transgenic tomato plants with uniform growth potential were selected. The first group served as the control; wild-type and transgenic tomato plants were treated using water. The second group served as the treatment group. Tomato plants were planted in 500 mL pots (matrix culture). Wild-type and *SlCOMT1* overexpression transgenic plants both growing five leaves were treated using 200 mL 800 mM NaCl twice a week at the same time. One week later, 0.1 g of different plant parts were harvested. Groups of three samples were pooled and used to measure the content of hydrogen peroxide, superoxide and proline. This experiment was repeated three times, and the results of three parallel experiments were averaged. The DPS data combing system and Tukey’s multiple comparison method (*p* < 0.05; *p* < 0.01) were used for statistical analysis.

## Figures and Tables

**Figure 1 molecules-24-01514-f001:**
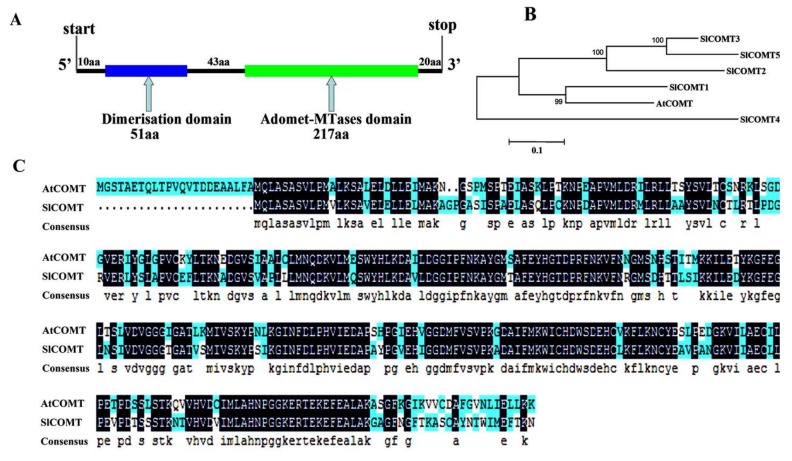
Bioinformatic analysis of the SlCOMT1 protein. (**A**) Two domains of the SlCOMT1 protein. (**B**) Phylogenetic tree was constructed using the five tomato COMT proteins and *Arabidopsis* AtCOMT protein. (**C**) Comparison of predicted SlCOMT1 protein sequence with AtCOMT. SlCOMT1 (XP_004235028.1), AtCOMT (NP_200227.1). Sl, *Solanum lycopersicum*; At, *Arabidopsis thaliana*.

**Figure 2 molecules-24-01514-f002:**
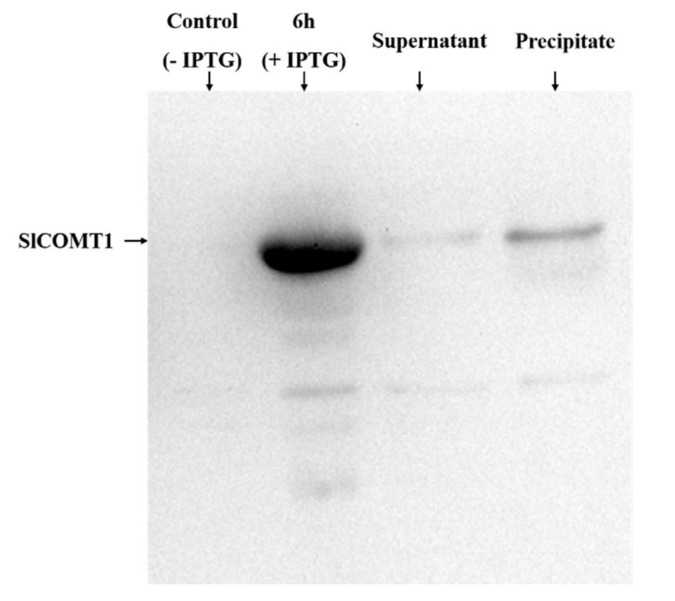
Induction of the SlCOMT1 protein in vitro. Lane 1, bacterial solution with no IPTG (control); lane 2, bacterial solution with IPTG cultured for 6 h at 37 °C; lane 3, supernatant derived from pET32a-SlCOMT cell lysate; and lane 4, precipitated SlCOMT1 protein.

**Figure 3 molecules-24-01514-f003:**
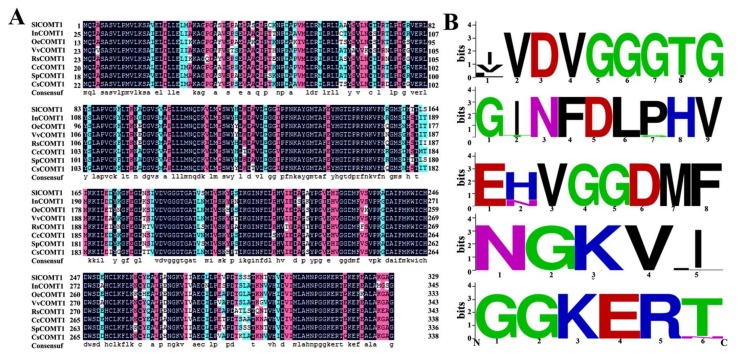
Evolution relationship of SlCOMT1 with other COMT proteins. (**A**) Comparison of amino acid sequences between tomato SlCOMT1 and COMT from other species, including *In*, *Ipomoea nil* (BAE94400.1); *Oe*, *Olea europaea* (XP_022844536.1); *Vv*, *Vitis vinifera* (XP_003634161.1); *Rs*, *Rauvolfia serpentine* (AOZ21153.1); *Cc*, *Capsicum chinense* (BAR88175.1)*; Sp*, *Solaum pennellii* (XP_015070697.1); and *Cs*, *Camellia sinensis* (ADN27527.1); (**B**) Five conserved domains of the COMT proteins.

**Figure 4 molecules-24-01514-f004:**
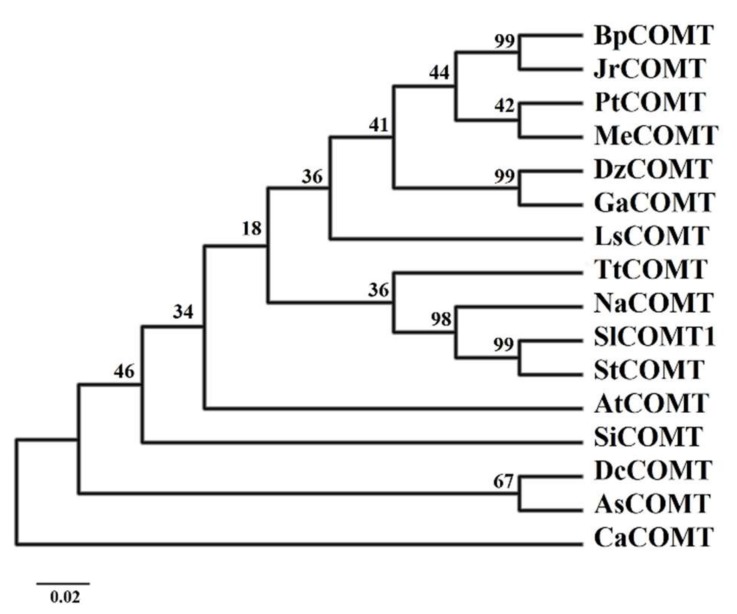
Phylogenetic analysis of the SlCOMT1 protein and its homologs. The proteins in the phylogenetic tree include *Bp*, *Betula pendula* (FJ667539.2); *Jr*, *Juglans regia* (XP_018828596.1); *Pt*, *Populus* Table 002321948. *Me*, *Manihot esculenta* (XP_021627291.1); *Dz, Durio zibethinus* (XP_022736469.1); *Ga, Gossypium arboretum* (XP_017611038.1); *Ls, Liquidambar styraciflua* (AF139533.1); *Tt*, *Thalictrum tuberosum* (AF064694.1); *Na*, *Nicotiana attenuata* (OIT03318.1); *St*, *Solanum tuberosum* (XP_015164331.1); *Si*, *Sesamum indicum* (XP_011075886.2); *Dc*, *Daucus carota* (XM_017381671.1); *As*, *Anthriscus sylvestris* (AB820126.1); and *Ca*, *Capsicum annuum* (NP_001311774.1).

**Figure 5 molecules-24-01514-f005:**
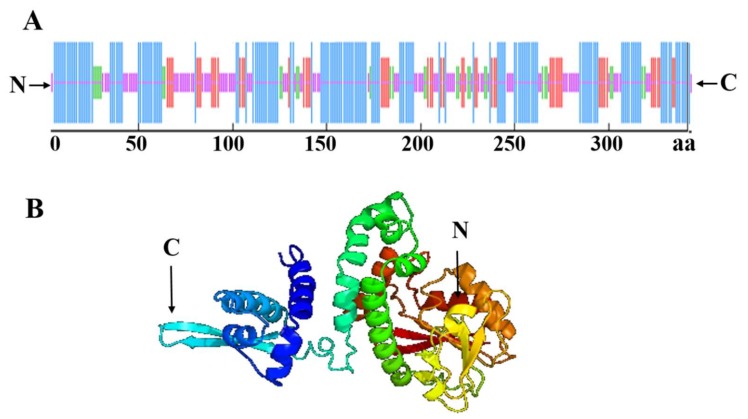
Structure prediction of the SlCOMT1 protein. (**A**) The secondary structure prediction of the SlCOMT1 protein. The blue indicates α-helices; purple indicates random coils; red indicates extended strands; green indicates β-turns; and the horizontal numbers indicate the positions of the amino acids. (**B**): The predicted three-dimensional structure of the SlCOMT1 protein.

**Figure 6 molecules-24-01514-f006:**
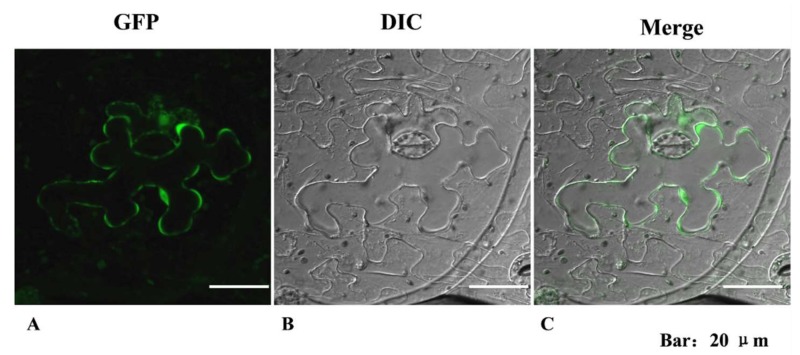
Localization of SlCOMT1. (**A**) Green fluorescence of SlCOMT1-PRI. (**B**) Bright-field image of *Agrobacterium*-infiltrated tobacco leaf. (**C**) The merged fluorescent images. The tobacco leaves were injected with the transgenic *Agrobacterium* liquid, and then cultivated in the culture chamber for 2 days and observed by confocal microscopy. GFP, Green Fluorescence Protein; DIC, Diascoptic Lighting Channel.

**Figure 7 molecules-24-01514-f007:**
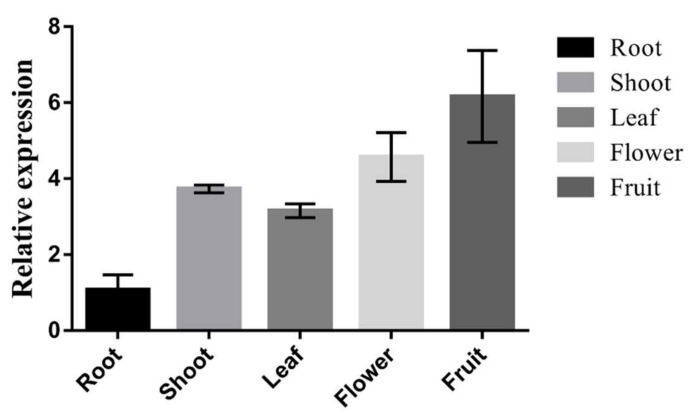
Expression levels of *SlCOMT1* in different tomato tissues.

**Figure 8 molecules-24-01514-f008:**
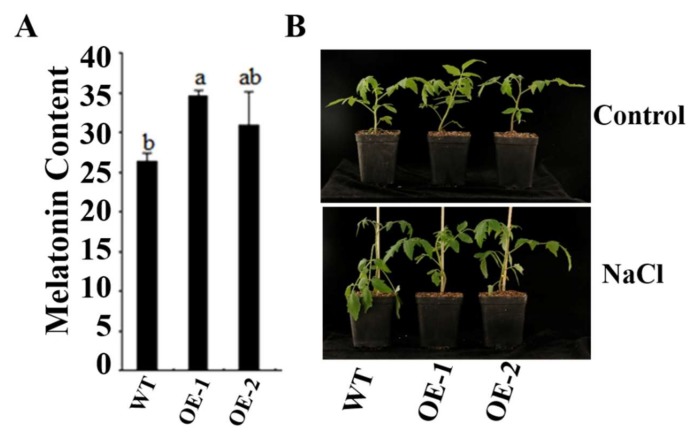
(**A**) Melatonin content in WT, OE-1, and OE-2 tomato plants. The same letter in the same growing season means no significant differences among three biological replicates (*p* < 0.05). Error bars represent standard error. (**B**) Growth status of WT, OE-1, and OE-2 after 800 mM NaCl treatment. WT, wild-type; OE, overexpression transgenic tomato.

**Figure 9 molecules-24-01514-f009:**
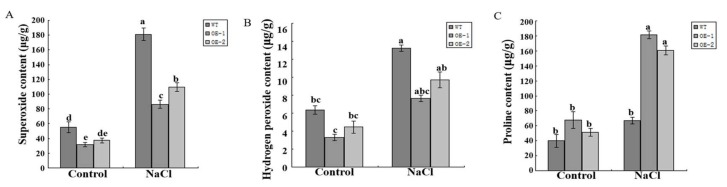
(**A**–**C**) were respectively the contents of superoxide, hydrogen peroxide, and proline in WT, OE-1, and OE-2 tomato plants, respectively. The same letter in the same growing season means no significant differences among three biological replicates (*p* < 0.05). Error bars represent SE.

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
