# Peer review of "Overexpression of the Melatonin Synthesis-Related Gene SlCOMT1 Improves the Resistance of Tomato to Salt Stress"

_molecules, 2019, doi:10.3390/molecules24081514_

Round 1

Reviewer 1 Report

The manuscript describes the transformation of tomato with the gene SlCOMT which is found to encode an enzyme belonging the enzyme family COMT, which has been reported to catalyze some reactions during melatonin biosynthesis. The authors also aimed at proving that the overexpression of the same gene alleviates salt stress (main title) in tomato. However, to eliminate speculations, I suggest the authors to find out what is the expression of the gene in both types of plants (normal and transformed) under salt stress and to compare it with the control plants. Furthermore, the manuscript is in several places written in very poor English and is hard to me to study sentence by sentence what exactly the authors meant to say. These sentences often make no sense or are incomplete and are located in the following lines: 32-33; 34-35; 60-62; 205-207; 219-220; plus almost whole abstract needs rewriting - phrases such as "resistance form plants", "which highly expressed in fruits", "was found locating", "in tomato leaf positively related to", "showed significantly lower", "obviously" and "taken collectively" are not acceptable. In general, I strongly suggest the authors to consult a native English speaker to check the manuscript.

More specific points:

Keywords: it is quite unusual quoting the object species two times within the keywords. It is also unusual to quote this in plural (tomatoes)

L28: I'm not aware of the term "pine pine voxel". Neither is Google.

L30: only in mammals?

L33: phytomelatonin - one word, without dash and quotation marks.

L34: reference is lacking in the end of the sentence (...molecule.)

L43-45: please provide EC numbers for the ststed enzymes.

L47: after [24] there is redundant spaces and a comma instead of a full stop. Further write: "serotonin is converted..."

L51: instead of "cytoplasm of plants" write "plant cytoplasm".

L53: "On the basis of the type of enzymatic catalysis..."

L63-64: the stated characteristics are not specific just for tomato. Please rephrase.

L72: please add "potential" before "stress-"

L79-86: it is not very clear whether the authors compare proteins or coding sequences in the Figure 1 and throughout the corresponding text.

L94: "...of 5.74. The..."

L102: "...(Figure 3A). They..."

L110: Camellia sinensis in italics

L118: instead of "and both of two plants are from Solanaceae" write "since reported in two congeneric species".

L124: "attenuata".

L144: "localized" instead of "localizing".

L149: No need to describe the bars. They are already indicated in the figure.

L172-173: phrase "that similar to that under drough treatment" sounds erratic. Put a period here and start with: "In contrast, it seemed normal..."

L174: add "both" after "measured in".

L177: not all values in the graphs are significanly different between control and NaCl-grown plants.

L188-189: delete the sentence, but leave the references. The second sentence only repeats the statement from the first one.

L191: add "and" before "after".

L192-194: comparing proteins with genes?

L196-198: delete last two sentences. They are redundant.

L205: please explain how "proline accumulation can respond..."

L221-223: repeating.

L229: add species author name.

L232: "pots" instead of "pot".

L235: from which tissue the authors extracted DNA?

L240: "separated" instead of "detected".

L332: write: ""Two groups each made of both wild-type and transgenic..."

L325: how frequently and in which volume (solution/substrat) were the plants treated with 800 mM NaCl?

Author Response

Point 1 However, to eliminate speculations, I suggest the authors to find out what is the expression of the gene in both types of plants (normal and transformed) under salt stress and to compare it with the control plants. Furthermore, the manuscript is in several places written in very poor English and is hard to me to study sentence by sentence what exactly the authors meant to say. 

Response 1: COMT is a key melatonin biosynthesis-related gene, and it is reported to improve melatonin production and positively contributes to strengthen both biotic and abiotic resistances in plants (Li et al., 2016, Choi et al., 2017). COMT transcript level was generally increased under stress conditions (Wang et al., 2018, Yang et al., 2019), leading to more melatonin production. Therefore, COMT can be induced by biotic and abiotic stress in various plant species, and the stress resistance would enhance in response to higher melatonin concentration.

We regret that assays were not performed to detect the transcript level of SlCOMT1 both in wild-type and transgenic plants under salt stress, while it was increased in SlCOMT1 overexpression transgenic plants under normal condition. Sorry for failing to detect SlCOMT1 transcript level both in wild-type and transgenic plants under salt stress, because there are no seeds nor seedlings of SlCOMT1 transgenic tomatoes are available to perform the assay at present. In this study, SlCOMT1 was found to promote melatonin production in tomato, and to enhance the tomato resistance to salt stress. Importantly, the increased transcript level of SlCOMT1 correlates positively to the elevated melatonin content in genetically transformed tomato. Thus, previous reports and the data in this study would prove the expression of SlCOMT1 in both types of plants (normal and transformed) under salt stress.

Besides, the whole manuscript has been carefully checked both by English language edition company “EditorBar” and a native English speaker Lawrence Ji from University of Pittsburgh. We are hopeful that it would be satisfied with your requirement.

More specific points:

Point 2: Keywords: it is quite unusual quoting the object species two times within the keywords. It is also unusual to quote this in plural (tomatoes)

Response 2: The original keywords were changed to be “Genetical transformation”. (Line 25, in the revised version)

Point 3: I'm not aware of the term "pine pine voxel". Neither is Google.

Response 3:  “pine pine voxel” was changed to be “pineal hormone” in the revised version. (Line 28)

Point 4: only in mammals?

Response 4:  This sentence discussed melatonin functions in mammals, and related reference was listed in L371-389, its functions in plants were introduced from L32 to L42.

Point 5: L33: phytomelatonin - one word, without dash and quotation marks.

Response 5: phyto-melatonin was changed to “phytomelatonin” in manuscript. (Line 33, 39, 52, in the revised version )

Point 6: L34: reference is lacking in the end of the sentence (...molecule.)

Response 6:  Reference 8 was added in the revised version. (L34, L384-385, in the revised version)

Point 7:L43-45: please provide EC numbers for the stated enzymes.

Response 7:  EC number were provided for the enzymes mentioned in the manuscript. More specifically, tryptophan hydroxylase (TPH, EC 1.14.16.4), tryptophan decarboxylase (TDC, EC 4.1.1.28), tryptamine 5-hydroxylase (T5H, EC 1.1.13), serotonin N-acetyltransferase (SNAT, EC 2.3.1.87), N-acetylserotonin-O-methyltransferase (ASMT, EC 2.1.1.4), and caffeic acid O-methyltransferase (COMT, EC 2.1.1.68) (Line 44-47, in the revised version)

Point 8:L47: after [24] there is redundant spaces and a comma instead of a full stop. Further write: "serotonin is converted..."

Response 8:  In revised manuscript, a comma was replaced by a full stop, and the subsequent sentence was changed to “Afterwards, serotonin is converted into N-acetylserotonin...”.  (Line 48-49, in the revised version)

Point 9: L51: instead of "" write "plant cytoplasm".

Response 9: “cytoplasm of plants” was modified to be “plant cytoplasm”(Line 53)

Point 10: L53: "On the basis of the type of enzymatic catalysis..."

Response 10:  “the type of” were added in the revised version. (Line 55)

Point 11:L63-64: the stated characteristics are not specific just for tomato. Please rephrase.

Response 11: The original sentence was change to be “Tomato, one of the most highly consumed and extremely important horticultural plants, has been studied as a model plant for some aspects of plant growth and development.” in the revised version. (Line 64-65)

Point 12: L72: please add "potential" before "stress-"

Response 12:  “potential” were added in the revised version. (Line 73)

Point 13: L79-86: it is not very clear whether the authors compare proteins or coding sequences in the Figure 1 and throughout the corresponding text.

Response 13:  Sequences used in Figure 1 were the protein sequences, we have now described  it in a better way. (Line 81, 83 and 88-90, in the revised version)

Point 14: L94: "...of 5.74. The..."

Response 14:  “,” was modified to be “.”, and “the” was changed to be “The”. (Line 97, in the revised version)

Point 15: L102: "...(Figure 3A). They..."

Response 15:  “,” was modified to be “.” as your suggested. (Line 105, in the revised version)

Point 16: L110: Camellia sinensis in italics

Response 16:  Camellia sinensis was used in italics in the revised version. (Line 114)

Point 17: L118: instead of "and both of two plants are from Solanaceae" write "since reported in two congeneric species".

Response 17:  The original sentence was modified to be “since reported in two congeneric species” in the revised version. (Line 120-121)

Point 18: L124: "attenuata".

Response 18:  "attenuate" was replaced by "attenuata" in the revised version. (Line 127)

Point 19: L144: "localized" instead of "localizing".

Response 19:  " localizing " was replaced by " localized " in the revised version. (Line 146)

Point 20: L149: No need to describe the bars. They are already indicated in the figure.

Response 20: Bars, 20 μm.” was deleted in the revised version. (Line 151)

Point 21: L172-173: phrase "that similar to that under drought treatment" sounds erratic. Put a period here and start with: "In contrast, it seemed normal..."

Response 21:  “that similar to that under drought treatment” was deleted and start with " In contrast, the leaves looked normal and healthy in transgenic plants." in the revised manuscript. (Line 175-176)

Point 22: L174: add "both" after "measured in".

Response 22:  "both" was added after "measured in" in the revised version. (Line 177)

Point 23: L177: not all values in the graphs are significantly different between control and NaCl-grown plants.

Response 23:  Yes, some values are not significantly different between control and NaCl-grown plants. Under normal development, the levels of superoxide, hydrogen peroxide and proline varied slightly both in wild-type and transgenic tomato plants (Figure 9A-C). Compared with control plants, the levels of superoxide and hydrogen peroxide increased both in wild-type and transgenic tomato plants under treatment using 800 mM NaCl, but they were lower in transgenic plants (Figure 9A-B). On the contrary, the level of proline was significantly higher in transgenic plants than in control plants (Figure 9C).

  We have already modified these descriptions in the revised version. (Line 178-183)

Point 24: L188-189: delete the sentence, but leave the references. The second sentence only repeats the statement from the first one.

Response 24:  The sentence was deleted in the revised version. (Line 192)

Point 25: L191: add "and" before "after".

Response 25: The original sentence was changed to be “and the melatonin synthesis-related gene SlCOMT1 was isolated based on homology comparison using Arabidopsis AtCOMT protein and the five tomato COMT proteins” in the revised version.(Line 194-195)

Point 26: L192-194: comparing proteins with genes?

Response 26:  In this study, five COMT homologous genes were identified in the tomato genome, and the melatonin synthesis-related gene SlCOMT1 was isolated based on homology comparison using Arabidopsis AtCOMT protein and the five tomato COMT proteins. We discussed it in a better way in the manuscript. (Line 193- 195)

Point 27: L196-198: delete last two sentences. They are redundant.

Response 27: The last two sentences were deleted in the revised version. (Line 197)

Point 28: L205: please explain how "proline accumulation can respond..."

Response 28: During normal cellular metabolism, ROS are generated by oxidative reaction process of mitochondrial respiration and photosynthesis process, and they act as signaling molecules during cellular repair processes at low amounts (He et al., 2017). Once the plant is under environment stresses, its cells simultaneously initiate a series of response mechanisms and stress signals, such as the activation of cellular ROS scavenging mechanisms, which can trigger the production of reactive oxygen scavenging enzymes and antioxidants, including POD and SOD which work on scavenging excessive ROS, thereby alleviating or eliminating oxidative stress (Miller et al., 2010). Under salt stress, the dynamic equilibrium of the production and elimination of reactive oxygen species in plant cells is disrupted, thereby causing the production of superoxide. Therefore, onset of cellular oxidative damage is the hallmark of salt stress, which is indicated by levels of superoxide and hydrogen peroxide (Martinez et al., 2018). Furthermore, proline in small amount plays multiple roles, such as stabilization of membrane and proteins, redox homeostasis and regulation of salt stress-responsive genes expression (Carillo, 2018, Ferchichi et al., 2018). Superoxide, hydrogen peroxide and proline contents can respond to many environmental stresses in plants, including salt stress, and the accumulation of hydrogen peroxide and superoxide can disrupt the dynamic balance of cells under environmental stress (Neill et al., 2002; Claussen, 2005; Wahid et al., 2007). Exogenous melatonin could have helped the tomato plants to bear the environmental stress by regulating the antioxidant system, proline and carbohydrates metabolism (Manzer et al., 2019). In this study, under treatment using NaCl, SlCOMT1 overexpression transgenic plants displayed the increased proline and the decreased hydrogen peroxide and superoxide levels, which were resulted from the reduced oxidative damage by extra melatonin that can scavenge ROS in plant cells. As a result, melatonin produced by the SlCOMT1 overexpression improved the growth characteristics of tomato compared to wild-type plants.

    We discussed it in a better way and added the above in the revised version. (Line 198-218)

Point 29: L221-223: repeating.

Response 29:  The original sentence was modified be “Additionally, SlCOMT1 overexpression transgenic tomato plants enhanced the resistance to salt stress.(Line 228-229)

Point 30: L229: add species author name.

Response 30:  Tomato (S. lycopersicum  Mill. cv. Ailsa Craig) was used for generating transgenic plant. (Line 243)

Point 31: L232: "pots" instead of "pot".

Response 31:  "pot" was replace by “pots” in the revised version. (Line 247)

Point 32: L235: from which tissue the authors extracted DNA?

Response 32:  RNA not DNA was extracted from tomato leaves, and we have added it in the L249. (in the revised version)

Point 33: L240: "separated" instead of "detected".

Response 33: "detected" was changed to be "separated" in the revised version. (Line 254)

Point 34: L322: write: ""Two groups each made of both wild-type and transgenic..."

Response 34: “made” and “both” were added in the manuscript. (Line 346)

Point 35: L325: how frequently and in which volume (solution/substrate) were the plants treated with 800 mM NaCl?

Response 35: OMT1 overexpression transgenic plants both growing five leaves were treated using 200mL 800mM NaCl twice a week at the same time. One week later, samples were pooled and used to measure the content of hydrogen peroxide, superoxide and proline. We have already added the method in the manuscript. (Line 349-352)

References

Carillo P. GABA Shunt in Durum Wheat. Front. Plant Sci. 2018, 9: 100.

Claussen, W. Proline as a measure of stress in tomato plants. Plant Sci. 2005, 168: 241-248.

Choi GHLee HYBack K. Chloroplast overexpression of rice caffeic acid O-methyltransferase increases melatonin production in chloroplasts via the 5-methoxytryptamine pathway in transgenic rice plants. J. Pineal Res. 2017, 63 (1). 

Ferchichi S, Hessini K, Dell’Aversana E, D’Amelia L, Woodrow P, Ciarmiello LF, Fuggi A, Carillo P.  Hordeum vulgare and Hordeum maritimum respond to extended salinity stress displaying different temporal accumulation pattern of metabolites. Funct. Plant Biol. 2018, 45:1096–1109. 

He L, He T, Farrar S, Ji LB, Liu TY, Ma X. Antioxidants maintain cellular redox homeostasis by elimination of reactive oxygen species. Cell Physiol. Biochem. 2017, 44:532–553.

Li W, Lu J, Lu K, Yuan J, Huang J, Du H, Li J. Cloning and Phylogenetic Analysis of Brassica napus L. Caffeic Acid O-Methyltransferase 1 Gene Family and Its Expression Pattern under Drought Stress. PLoS One. 2016,11(11): e0165975.

Manzer HS, Saud, A, Mutahhar Y, Al-Khaishany, MY, Khan MN, Abdullah A, Hayssam MA, Ibrahim AA, Abdulaziz AA. Exogenous melatonin counteracts NaCl-induced damage by regulating the antioxidant system, proline and carbohydrates metabolism in tomato seedlings. Int. J. Mol. Sci. 2019, 20: 353.

Martinez V, Nieves-Cordones M, Rodenas, R, Mestre TC, Garcia-Sanchez F, Rubio F, Nortes PA, Mittler R, Rivero RM. Tolerance to stress combination in tomato plants: New insights in the protective role of melatonin. Molecules 2018, 23: 535.

Miller G, Suzuki N, Ciftci S, Mittler R. Reactive oxygen species homeostasis and signalling during drought and salinity stresses. Plant Cell & Environ. 2010, 33: 453-467.

Neill, S.J.; Desikan, R.; Clarke, A.; Hurst, R.D.; Hancock, J.T. Hydrogen peroxide and nitric oxide as signalling molecules in plants. J. Exp. Bot. 2002, 53:1237-1247.

Yang WJDu YTZhou YBChen JXu ZSMa YZChen MMin DH. Overexpression of TaCOMT improves melatonin production and enhances drought tolerance in transgenic Arabidopsis. Int. J. Mol. Sci. 2019, 20(3): 652.

Wahid, A.; Gelani, S.; Ashraf, M.; Foolad, M.R. Heat tolerance in plants: An overview. Environ. Exp. Bot. 2007, 61:199-223.

Wang MZhu XWang KLu CLuo MShan TZhang Z. A wheat caffeic acid 3-O-methyltransferase TaCOMT-3D positively contributes to both resistance to sharp eyespot disease and stem mechanical strength. Sci. Rep. 2018, 25,8(1):6543.

Reviewer 2 Report

In this study, the authors investigated melatonin biosynthesis-related SlCOMT1 gene was cloned from tomato (Solanum lycopersicum cv. Ailsa Craig), which highly expressed in fruits. Melatonin content in SlCOMT1 overexpressing transgenic tomato plants was significantly higher than that in wild-type plant. SOD and H2O2 levels showed significantly lower, while the content of proline was obviously higher in SlCOMT1transgenic tomatoes. SlCOMT1 closely associated with melatonin biosynthesis confers the significant salt tolerance, providing a clue to cope with the growing global salination in agricultural production.

The topic is within the scope of this journal.  However, the manuscript preparation does not reach to the standards of scientific publication.

English of the MS needs to be greatly improved. The English of the whole article has to be checked carefully to eliminate linguistic errors.

·         In introduction and discussion part author needs to cite more recent references.

·  Overexpression technique they used Agrobacterium mediated genetic transformation. Authors need to provide tomato transformation (Stages of transformation and molecular identification of tomato) transgenic pictures (transgenic callus initiation, shoot regeneration, rooting and hardening of transgenic plants).

·         Authors need to provide detailed conclusion.

Author Response

Point 1: English of the MS needs to be greatly improved. The English of the whole article has to be checked carefully to eliminate linguistic errors.

Response 1The whole manuscript has been carefully checked both by English language edition company “EditorBar” and a native English speaker Lawrence Ji from University of Pittsburgh. We are hopeful that it would be satisfied with your requirement.

Point 2: In introduction and discussion part author needs to cite more recent references.

Response 2More recent references were added in the introduction and discussion part in the revised version, such as in L63 (Reference no.29, 30), L65 (Reference no.31) and L198 (Reference no.38), L207 (Reference no.40), L209 (Reference no.41, 42).

Point 3: Overexpression technique they used Agrobacterium mediated genetic transformation. Authors need to provide tomato transformation (Stages of transformation and molecular identification of tomato) transgenic pictures (transgenic callus initiation, shoot regeneration, rooting and hardening of transgenic plants).

Response 3Tomato transformation was performed using cotyledons at the cotyledons stages (Line 302-303, in the revised version). Molecular identification of transgenic tomato were performed by PCR and quantitative RT-PCR. To identify transgenic tomatoesthe cDNA of both wild-type and transgenic plants were used as a template to detect the expression level of SlCOMT1, the plasmid DNA of SlCOMT1 was used as positive control, and ddH2O as negative control. The transgenic samples with increased SlCOMT1 expression level were used for further study. Transcript level of SlCOMT1 was detected using quantitative RT-PCR. Primers for quantitative RT-PCR (Supplementary Table 1) were used to detect the expression level of SlCOMT1. Quantitative RT-PCR reaction conditions were listed as supplementary Table 4, with 30 cycles for fluorescence collection from denaturation to extension, and finally a quantitative analysis by 2-∆∆CT method.

We have included the above in Materials and Methods (Line 324-333, in the revised version). Additionally, some transgenic pictures were also supplemented as supplementary Figure 3, and the related descriptions were added in L303-314 in the revised version.

Point 4: Authors need to provide detailed conclusion.

Response 4In conclusion, the present study shows that melatonin biosynthesis-related gene SlCOMT1 isolated from tomato is localized in the cytoplasm, and is highly expressed in fruits. Melatonin content in SlCOMT1 overexpression transgenic tomato plants is significantly higher than that in wild-type plants. The transgenic plants display increased proline levels and decreased hydrogen peroxide and superoxide levels, and the transgenic tomatoes tolerated salt stress better than the wild-type tomatoes. The results indicate that SlCOMT1 is closely relate to melatonin production and functions in the improvement of plant resistance to abiotic stress.

We added conclusions in the discussion section. (Line 233-240, in the revised version)

Round 2

Reviewer 1 Report

Line 148 - "leaf" instead of "leave".

Reviewer 2 Report

Requested corrections were carried out by authors.